# Brain Changes in Alcohol Induced Liver Cirrhosis Patients: Insights from Quantitative Susceptibility Mapping

**DOI:** 10.3390/biomedicines13122937

**Published:** 2025-11-29

**Authors:** Andrej Vovk, Stefan Ropele, Sebastian Stefanovic, Borut Stabuc, Dusan Suput, Marjana Turk Jerovsek, Gasper Zupan

**Affiliations:** 1Center for Clinical Physiology, Faculty of Medicine, University of Ljubljana, Zaloska 4, 1000 Ljubljana, Slovenia; 2Neuroimaging Research Unit, Department of Neurology, Medical University of Graz, Auenbruggerplatz 22, 8036 Graz, Austria; stefan.ropele@medunigraz.at; 3Diagnostic Center Bled Group, Pod Skalo 4, 4260 Bled, Slovenia; 4Department of Gastroenterology and Hepatology, University Medical Center Ljubljana, Japljeva ulica 2, 1000 Ljubljana, Slovenia; 5Institute of Oncology, Department of Radiology, Zaloska 2, 1000 Ljubljana, Slovenia

**Keywords:** hepatic encephalopathy, QSM, enlarged perivascular spaces

## Abstract

**Background and Purpose:** Hepatic encephalopathy (HE) is a neuropsychiatric syndrome associated with liver cirrhosis (LC) that often results in cognitive impairment. Minimal HE (mHE), a subtle form of the condition, significantly affects patients’ quality of life. Advanced imaging techniques, such as quantitative susceptibility mapping (QSM), provide new insights into the brain changes associated with HE. **Materials and Methods:** The study included 28 patients (17 with mHE and 11 without) with alcohol-induced LC and 25 healthy controls. MR imaging, including QSM, was utilized to assess microstructural tissue changes and iron deposition in the brain. Cognitive function was assessed through a neuropsychological test battery. QSM quantified magnetic susceptibility in deep gray matter, while enlarged perivascular spaces (EPVS) were evaluated using T2-weighted images. Statistical analyses, including non-parametric tests and linear regression, assessed differences in susceptibility and their correlation with cognitive performance and EPVS. **Results:** Significant differences in cognitive performance and brain susceptibility were observed between patients and controls. Patients exhibited lower susceptibility in the caudate nucleus with the accumbens (CNA); mHE patients, in particular, had a significant reduction in CNA susceptibility. Additionally, EPVS grade correlated positively with cognitive decline, suggesting that EPVS may play an essential role in the pathophysiology of mHE. **Conclusions:** This study demonstrates that QSM can detect subtle brain changes in LC patients, with decreased susceptibility in the CN (caudate nucleus) linked to cognitive impairment in mHE. The role of EPVS in HE warrants further investigation, as it may affect the efficacy of current diagnostic and therapeutic approaches. These findings highlight the potential of QSM to improve HE assessment.

## 1. Introduction

Hepatic encephalopathy (HE) is a challenging and complex neuropsychiatric syndrome caused by liver dysfunction that affects millions of individuals worldwide. Characterized by a spectrum of cognitive, motor, and behavioral impairments, HE poses a significant clinical burden and affects both the quality of life and survival of patients with liver cirrhosis (LC) and other hepatic disorders. LC is an irreversible fibrosis of the liver [1]. In Europe, the most common cause of LC is chronic alcohol overconsumption. LC caused by other factors is relatively rare in clinical practice in Europe; for example, chronic hepatitis B affects only about 0.5% of the European population [2]. This enigmatic disorder lies at the intersection of gastroenterology, neurology, and psychiatry, making it a topic of utmost importance for clinicians, researchers, and healthcare systems [3].

HE arises primarily in the context of chronic liver disease, as impaired hepatic function leads to the accumulation of neurotoxic substances, especially ammonia, in the bloodstream. These toxins penetrate the blood–brain barrier, ultimately disrupting normal brain function and leading to a range of neurological and neuropsychiatric symptoms [4].

The manifestation of HE can range from subtle cognitive impairment, referred to as minimal HE (mHE), to severe and life-threatening hepatic coma. The clinical presentation is characterized by symptoms such as confusion, asterixis (flapping tremor), impaired attention, and altered consciousness, making it a challenging condition to diagnose and manage. Since HE is potentially reversible, recognition of the early stages of HE, such as mHE, is of most importance [5]. HE is associated with increased mortality in LC patients [6]. mHE is defined as HE without clinical symptoms; however, mild deficits in some cognitive domains can be assessed by neuropsychometric testing [3]. mHE has a high prevalence in patients with liver cirrhosis, ranging from 22% to 74% [3].

The striatum, comprising the CN and putamen, serves as the primary input center of the basal ganglia, which is closely connected with the substantia nigra and plays a crucial role in motor function regulation. Motor dysfunction is found in about half of patients with LC [7], and it is considered potentially reversible. The CN plays a critical role in visual stimulus–response learning and visual working memory functions [8]. Additionally, it is involved in a range of cognitive and affective processes, including learning, memory, reward, motivation, and emotion [9]. In LC, previous studies have found neuronal loss with spongiform degeneration, manganese deposition, and decreased cerebral blood flow [10] in the basal ganglia. Moreover, lower striatum volume, which correlates with decreased cognitive performance, has also been demonstrated in LC [11], highlighting the importance of basal ganglia involvement in the pathophysiology of LC. 

Enlarged perivascular spaces (EPVS) are dilated, fluid-filled intracranial spaces that run parallel to penetrating arterioles in the brain [12]. EPVS are believed to be an important component of the glymphatic system, which is responsible for clearing potentially toxic products from the brain parenchyma [13]. Studies have demonstrated a connection between high EPVS burden and neurodegenerative processes such as several types of dementia [13,14,15], cognitive decline [16], neurovascular diseases like cerebral amyloid angiopathy, microbleeds, and stroke [17], as well as normal aging and hypertension [18]. However, no study has investigated a possible link between EPVS and HE so far.

Quantitative susceptibility mapping (QSM) has proven to be a powerful tool for assessing the magnetic contributions of different constituents of brain tissue and their alterations. Magnetic susceptibility is mostly determined by the magnetic properties of tissue water, but can be shifted by paramagnetic contributions from iron and diamagnetic contributions from calcium and the lipids and proteins of myelin [19]. Therefore, QSM has the potential to assess disease-induced changes in these components that remain undetected with conventional MRI.

This work aims to evaluate disease-induced changes in EPVS and magnetic susceptibility in the brains of HE patients to assess the relevance of morphological and magnetic features for evaluating liver cirrhosis-induced brain changes. Our findings leverage the unique capabilities of QSM to investigate the spatial distribution and potential pathological correlates of diamagnetic and paramagnetic deposits in the brain. Ultimately, this study will contribute to a deeper understanding of the pathophysiology of this complex neurological disorder.

## 2. Materials and Methods

**Subjects:** 34 patients diagnosed with LC based on clinical criteria and caused by excessive alcohol consumption, without known hepatitis B or C infection, and with no other primary gastrointestinal (GI) disease, were included in this study. 28 age- and gender-matched healthy controls (HC) with no GI, neurological, or psychiatric complaints were also invited. Due to data acquisition limitations and segmentation issues, the final group size was 28 patients and 25 HCs. Patients with LC were recruited at the outpatient office of the Department of Gastroenterology, University Medical Center. Patients were assessed by an attending gastroenterologist according to the Child-Pugh score, and possible concurrent HE was diagnosed clinically according to the standard West-Haven classification. Only patients with no evidence of overt HE (West-Haven 0) were included in the study. Patients were not treated for HE with medication, as there were no overt clinical signs of HE. Venous blood samples were obtained from the patients to determine ammonia levels on the day of MRI. After the MRI, all subjects underwent a series of neuropsychological tests. The battery of tests was designed to examine executive functions, concentration, visuospatial abilities, motor functions, and, to a lesser extent, memory. The test battery was adapted from the Psychometric Hepatic Encephalopathy Score [20] and consisted of standard tests in the following order: Numerical Connectivity Test A, Numerical Connectivity Test B, Serial Dot Test, Digit Symbol Test, and Line Tracing Test. Due to the expected limited computer skills of some participants, the tests were administered using paper and pencil. Time to completion (TTC) was recorded when a participant successfully completed all tests. Patients were divided into non-mHE and mHE groups. The inclusion criterion for the mHE group was a TTC > 2 standard deviations from the mean TTC of the HCs. TTC scores were not adjusted for participants’ age or educational level. Only patients without clinical evidence of vitamin B12 deficiency were included. Moreover, subjects with radiological findings suggestive of deficiency—including hyperintense T2 signal in the dorsal columns or any abnormal T2 hyperintensity within the deep white matter—were excluded.

The study was approved by the National Medical Ethics Committee, and informed consent was given by all subjects.

**MRI:** MRI of the brain was performed on a Philips Achieva 3.0T TX scanner (Philips Healthcare, Best, The Netherlands) equipped with a 32-channel head coil. The MR imaging protocol included several pulse sequences: a three-dimensional T1-weighted gradient echo sequence and a T2-weighted 3D turbo spin echo sequence, both with an isotropic voxel size of 0.7 mm, as well as a diffusion-weighted spin echo sequence with EPI readout and an isotropic resolution of 2 mm.

QSM data were obtained using a multi-echo gradient echo sequence with an isotropic resolution of 1 mm, 6 echoes, a first echo at 5.5 ms, and an echo spacing of 5 ms. Further details regarding the MRI sequence parameters are available in Section B.1.

**Image processing:** Susceptibility maps were computed from the multi-echo gradient echo sequence using a TGV-based algorithm [21], which provides high robustness by incorporating phase unwrapping, background field removal, and dipole inversion in a single iteration. By using the local field maps as input, the TGV algorithm addresses the limitations of the classical QSM approach by formulating QSM reconstruction as a single variational optimization problem. The algorithm minimizes a cost function that balances two terms: a data fidelity term, which ensures that the estimated susceptibility distribution, when convolved with the dipole kernel, matches the measured local field; and a regularization term, which encourages spatial smoothness in the susceptibility map while allowing for both sharp edges (at tissue boundaries) and gradual changes (within homogeneous tissues). In this work, we used the latest version of the algorithm available at https://www.neuroimaging.at/pages/qsm.php (accessed on 26 November 2025) [21]. The concentration of diamagnetic and paramagnetic components was quantitatively determined using a method that measures absolute magnetic susceptibility.

Our analysis focused on the deep gray matter regions, as elevated iron levels have previously been found in these areas in cirrhotic patients. We did not assess susceptibility changes in white matter because the counteracting effects of myelin and its orientational dependency hinder clear interpretation. Several methods have been proposed for segmentation of the basal ganglia and thalamus. We employed the multimodal segmentation technique MIST to improve the robustness of the segmentations [22]. MIST combines T1 and T2 contrast and fractional anisotropy (FA) maps to model intensity profiles in multiple images around the boundaries of the structure after nonlinear registration. The advantage of this approach is more robust segmentation, especially in regions where T1 contrast is low. FA maps were calculated from DWI data using the TORTOISE tools [23]. The TORTOISE tool “diff_prep” was applied, using quadratic eddy current correction, ANTSSyN EPI distortion correction, and noise reduction for registration and Gibbs ringing correction. DTI-related metrics, including MD, RD, and FA, were calculated using the tool “diff_calc” from TORTOISE. We focused on subcortical structures involved in cognition, emotion, and motor function, including the globus pallidus, amygdala, CNA, putamen, and thalamus. Graphical presentation of observed regions and their functional involvement is presented in Appendix A. The mean susceptibility of each region of interest (ROI) was calculated using AFNI’s “3dROIstats” program and averaged for both hemispheres [24].

EPVS in the basal ganglia, thalami, and internal capsule was assessed using T2-weighted images. EPVS was defined according to the Standards for Reporting Vascular Changes on Neuroimaging [25,26].

The quantification of EPVS began by identifying them as fluid-filled spaces that follow the path of vessels and exhibit a CSF-like signal. To differentiate them from lacunes, these spaces must not have a surrounding T2-hyperintense rim. National and European board-certified radiologist G.Z. visually reviewed both hemispheres, and the highest unilateral count was used. Subjects were categorized into groups with non-to-mild or moderate-to-high EPVS. A cut-off point was established by adapting the criteria from Potter et al. [27]. More than ten EPVS were defined as the threshold for the moderate-to-high group.

**Statistics:** Statistical analyses were performed using R (ver.4.1.2) [28] on the QSM values for selected subcortical regions, delineated by MIST, to identify significant differences or trends related to these regions. In addition, susceptibility values were filtered for outliers that occurred outside the 1.5-fold interquartile range above the upper quartile and below the lower quartile in each region to exclude possible measurement or recording errors. Pearson’s Chi-square test and the Wilcoxon rank sum test were used to test for statistical differences in the summarized data. The use of a complex linear mixed-effects model was considered, and its appropriateness was evaluated given the small group sizes. Complex models, particularly those with random effects, require a sufficient number of observations to ensure reliable and stable parameter estimates. To avoid issues of overfitting and potential instability, we employed a model selection approach based on the Akaike Information Criterion (AIC). More technical details regarding the AIC model selection are explained in Section B.2.

Based on the AIC comparison, the final model selected for the analysis was a mixed-effects model with the form: susceptibility ~ Group + (1 | SubID). This model includes a fixed effect for Group and a random intercept for each subject. The random effect was included to account for the non-independence of repeated measures, as data from both the left and right brain structures were included in the analysis. This approach ensured that the model properly handled the nested data structure while maintaining a parsimonious fit. To further explore and show dependencies of magnetic susceptibility on age or TTC, additional linear regression analyses were performed for each group in each region (susceptibility ~ Age + (1 | SubID) and susceptibility ~ TTC + (1 | SubID)).

## 3. Results

The general characteristics of patients and controls are summarized in Table 1, including gender, age, Mini-Mental State Examination (MMSE) score, time to complete the cognitive test, EPVS grade, CHILD rate, and distribution of HE subgroups.

Both groups were comparable in age, gender, and EPVS grade. However, we found statistically significant differences in cognitive performance between patients and HC, which was evident in the TTC. There was no statistically significant difference in MMSE scores or EPVS grading between the HC and patient groups. Among the HE subgroups, 61% of HE patients were diagnosed with mHE.

In patients, we found overall lower magnetic susceptibility in the basal ganglia and thalamus compared to HCs. The mixed-effect model indicates that subjects in the patients_minHE group have significantly lower susceptibility values in the CNA compared to the reference Control group (est. = −0.010732, std.err. = 0.003565, df = 52.4, *p* = 0.00401), while there is no significant difference for the patients_nonHE group. There is a non-significant trend for the patients_minHE group to have lower susceptibility values in the thalamus than the reference Control group, but it does not reach the typical threshold for statistical significance (est. = −0.0033442, std.err. = 0.0017415, df = 47.3, *p* = 0.0609) (Figure 1).

The results of the linear regression analysis of age and magnetic susceptibility in the CNA and putamen are presented in Figure 2. Statistically significant positive associations were found for HC in the CNA and the putamen (*p* < 0.05). In contrast, no significant trend was observed in the patients.

Two exemplary axial T2-weighted images at the basal ganglia level are shown in Figure 3 to demonstrate the spectrum of EPVS appearance. The result of the visual EPVS rating is summarized in Figure 4.

Visual rating revealed a higher prevalence of EPVS in patients compared to controls in commonly affected regions (putamen: 6 vs. 7.2, globus pallidus: 5.6 vs. 6.6, and internal capsule: 3.2 vs. 3.6). Additionally, more regions were affected in the patients. In the HC group, no EPVS were found in the thalamus or caudate nucleus.

Figure 5 shows how the presence of EPVS affected regional susceptibility. Except for the putamen, high to moderate EPVS was associated with lower magnetic susceptibility. However, there is a non-significant trend for the mod-highEPVS group to have lower susceptibility than the reference non-mildEPVS group in the thalamus, but it does not reach the typical threshold for statistical significance (Est. = −0.002748, std.err. = 0.001557, df = 48.7, *p* = 0.0839)

Magnetic susceptibility was also associated with the TTC of the cognitive tests, with a longer TTC related to lower susceptibility. However, a significant correlation was found only in the CNA of HE patients (Figure 6).

A statistically significant negative trend in susceptibility to TTC in the CNA was observed.

Further analysis was conducted to investigate the relationship between TTC and age. The results of the linear regression are shown in Figure 7.

Figure 7 highlights a strong positive association between TTC and EPVS for patients with EPVS (slope = 18.2, *p* < 0.001).

## 4. Discussion

In this study, changes in brain susceptibility in LC patients were quantified using QSM. The findings revealed lower susceptibility in the CNA and other deep gray matter nuclei of patients with mHE compared to HC, with no significant difference between non-mHE patients and HC. The striatum, which includes the CN and the putamen, serves as the primary input center of the basal ganglia, is closely connected with the substantia nigra, and plays a crucial role in motor function regulation. The CN is subdivided into the head, body, and tail, with the latter two particularly important for visual stimulus-response learning and visual working memory functions [8]. The CN is also involved in learning, memory, reward, motivation, and emotion [9].

T1 hyperintensity is often observed in the basal ganglia due to manganese accumulation in severe LC, although these changes can also occur in the substantia nigra, internal capsules, and cerebellar peduncles. This finding can be observed with no clinical signs as well. Occasionally, T2/FLAIR hyperintensities occur in the cortex and white matter, likely due to edema [29]. Detailed structural analyses have shown reduced CN volume in LC patients [30], with a significant reduction seen only in high-grade LC [11], suggesting that the neurodegenerative process in LC affects the basal ganglia.

Wang et al. [31] reported increased average susceptibility values in bilateral CN, indicating that iron deposition may be part of the pathogenesis in LC, with or without minimal HE. They also found a positive correlation between CN susceptibility values and performance on the Number Connection Test A, suggesting that iron deposition could induce oxidative stress, leading to neurodegeneration and cognitive impairment. However, it is important to note that their study did not use automated brain segmentation, and the ROIs were manually selected by radiologists, potentially limiting the analysis to only parts of the observed structures. Their study also reported unilaterally increased susceptibility values in the substantia nigra, thalami, and hippocampi, which we found unexpected given the systemic nature of LC pathogenesis.

Xia et al. [32] also reported increased susceptibility in the bilateral CN, but found decreased susceptibility in the right dentate nucleus. Similar to Wang et al., they observed unilaterally increased susceptibility in the substantia nigra and thalami. A key distinction from our study is the patient population: Wang et al. and Xia et al. primarily studied patients with hepatitis B-related LC, whereas our study focused on patients with alcohol-induced LC. Additionally, methodological heterogeneity was present across the studies. Such differences should be taken into account when interpreting and comparing the findings, as they may reflect variations in the underlying pathophysiological mechanisms.

Our analysis showed decreased CNA susceptibility in mHE patients compared to HC, with no differences between non-mHE patients and HC. In addition, a significant positive correlation was found between EPVS grade and TTC, suggesting that EPVS may play an important role in the pathophysiology of HE. The pathogenesis of HE is multifactorial, with hyperammonemia as a major contributing factor [33]. In LC, the damaged liver is unable to convert ammonia into urea, leading to ammonia accumulation in the brain, where it is metabolized only in astrocytes [34]. In advanced LC, elevated levels of ammonia, cytokines, and bile acids in the blood can accumulate in the brain along with a leaky blood–brain barrier (BBB) and trigger a harmful neuroinflammatory response. This can impair the function of the neurogliovascular unit, which includes afferent vessels, glial cells, the BBB, and the glymphatic pathway responsible for clearing of toxic compounds from the brain [35]. Afferent and efferent penetrating brain vessels are surrounded by PVS as they branch out from the subarachnoid space. Microscopy studies have shown that the penetrating arterioles are encased in a layer of pia mater that, together with the basement membranes of the astrocytes, forms an EPVS filled with CSF [36]. The PVS is a key component of the glymphatic system, which is responsible for clearing of potentially harmful products from the brain parenchyma. The exact mechanism of the glymphatic system is not yet entirely understood, but it is believed that the pulsation of penetrating arterioles drives CSF into the interstitial space of the glial cells and pushes waste proteins and other metabolic products into the perivenous space and subsequently into the meningeal lymphatic vessels and cervical lymph nodes, effectively clearing the brain parenchyma [13].

Hadjihambi et al. [37] provided evidence of glymphatic system dysfunction in a rat model of mHE. Using dynamic MR imaging, they demonstrated impaired gadolinium contrast agent penetration and clearance in the olfactory bulb and prefrontal cortex. They also found decreased expression of aquaporin-4 channels, which facilitate water flow between the periarterial space, interstitial space, and perivenous space in the olfactory bulb and prefrontal cortex, demonstrating two levels of glymphatic system impairment in mHE. Detection of EPVS by MR imaging depends on magnetic field strength and image resolution. At higher magnetic field strengths (≥3T), EPVS can be detected in most subjects, including healthy young adults and adolescents, in whom detection of EPVS may not be pathologic or have clinical implications [38].

Our results provide convincing evidence that the decrease in magnetic susceptibility is due to a partial volume effect with CSF in the enlarged EPVS. CSF has a susceptibility equal to weakly diamagnetic water (χ = −9.05 ppm) [39]. Consequently, in deep gray matter regions with high iron content and therefore higher magnetic susceptibility (χ normally up to +0.18 ppm), EPVS lead to a diamagnetic shift toward lower susceptibility values. Global assessment of these structures might also pick up small enlargements or perivascular spaces of small vessels that cannot be resolved with the limited resolution of conventional MRI. The ceiling of age-related iron accumulation in these structures and disease-induced acceleration usually occurs in the third to fourth decade of life, which could also explain why iron-related susceptibility changes might play only a minor role in our study. This becomes relevant when comparing our results with previous studies in which the included patients were much younger [31]. Integrating QSM into clinical workflows could potentially offer new insights into early diagnosis of HE; however, proper validation needs to be performed first.

Several limitations of the present study should be acknowledged. First, the control group was composed of healthy participants without a history of alcohol misuse. From a methodological perspective, the inclusion of a comparator group consisting of chronic alcohol consumers without any evidence of liver damage would have been preferable. However, such individuals, those who consume alcohol regularly yet exhibit no signs of hepatic injury, are relatively uncommon [40]. With a healthy control group, we cannot definitively separate the neurotoxic effects of chronic alcohol consumption from the neurobiological consequences of cirrhosis. However, the observed EPVS burden demonstrated a specific association with poorer cognitive performance (mHE) and distinct magnetic susceptibility changes in the Caudate Nucleus. This pattern, localized to the deep gray matter and correlated with current metabolic cognitive status, suggests that glymphatic dysfunction secondary to liver pathology is a significant contributing factor, distinct from the generalized cortical atrophy typically associated with alcohol use disorder alone. Second, the study was constrained by a relatively small sample size, which may have limited the statistical power to detect subtle differences between the groups and reduced the generalization of the findings. We acknowledge that the relatively small sample size is a limitation of this study. A larger sample would have provided greater statistical power to detect smaller differences between the groups. However, we believe the statistical significance observed for several key outcomes, even with the current sample size, highlights the presence of a meaningful effect. The significant findings indicate that the differences between the groups for these particular variables are likely genuine, as they were strong enough to be detected despite the lower statistical power. As an exploratory investigation, this study did not have a pre-determined effect size for power calculation. However, a post hoc power analysis based on our observed effect size (Cohen’s d = 0.9) reveals that the study had a power of 76% to detect an effect of this magnitude. This suggests our significant findings are likely robust. Our results, therefore, serve as a crucial preliminary finding, providing a strong rationale for future studies with a larger sample size. Such a study would have the necessary power to investigate whether the groups also differ on outcomes where no significant difference was found in the current study. We should also acknowledge that limiting our study to alcohol-induced LC may restrict the extrapolation of our findings to other etiologies of cirrhosis. However, given the high prevalence of alcohol-induced LC in the European demographic, our findings provide a significant contribution to understanding the neurocognitive manifestations specific to this patient group. Future studies should aim to validate these findings in patients with different etiologies of cirrhosis.

## 5. Conclusions

EPVS, though common in normal aging, is more prevalent in patients with minimal HE and is associated with poorer cognitive performance and lower magnetic susceptibility, particularly in the CNA. Although EPVS is linked to blood–brain-barrier disruption and glymphatic dysfunction, its role in mHE pathophysiology remains unclear. Our study demonstrates that QSM enables a global and unbiased assessment of perivascular changes in deep gray matter in mHE, providing a crucial preliminary finding and a strong rationale for future, larger-scale studies adequately powered to validate these initial observations and investigate outcomes lacking significant differences in the current analysis.

## Figures and Tables

**Figure 1 biomedicines-13-02937-f001:**
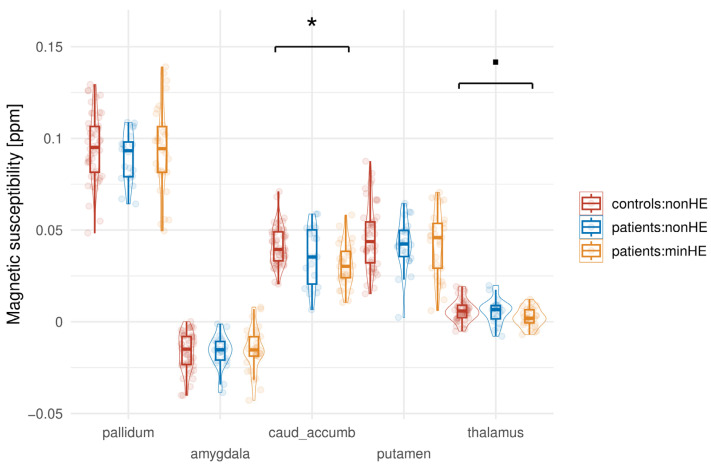
Magnetic susceptibility in subcortical brain regions, stratified by patients and HC. The line and the boxes represent the mean and standard deviation, respectively. Statistically significant differences between groups are indicated by * (*p* < 0.05), while a non-significant trend is indicated by ▪ (*p* = 0.06).

**Figure 2 biomedicines-13-02937-f002:**
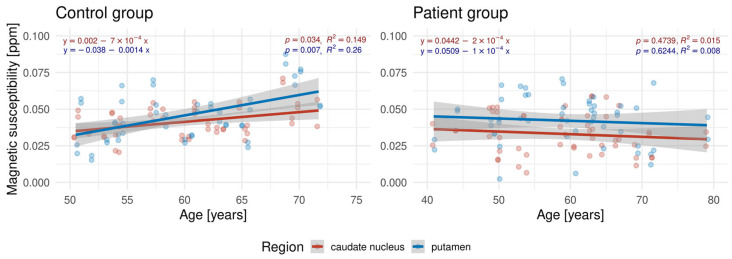
Age dependence of magnetic susceptibility in deep gray matter structures. The data were stratified by control and patient groups; presented on the left and right panels, respectively.

**Figure 3 biomedicines-13-02937-f003:**
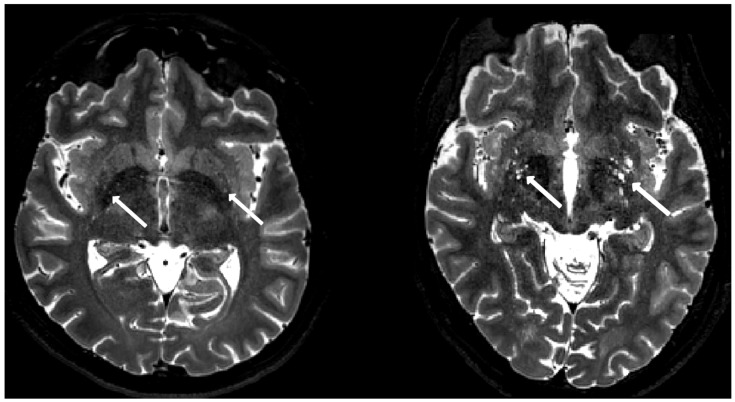
Examples of mild EPVS from a control subject (**left**) and high EPVS from a patient subject (**right**) are presented on a T2-weighted image. EPVS in the basal ganglia are indicated by arrows.

**Figure 4 biomedicines-13-02937-f004:**
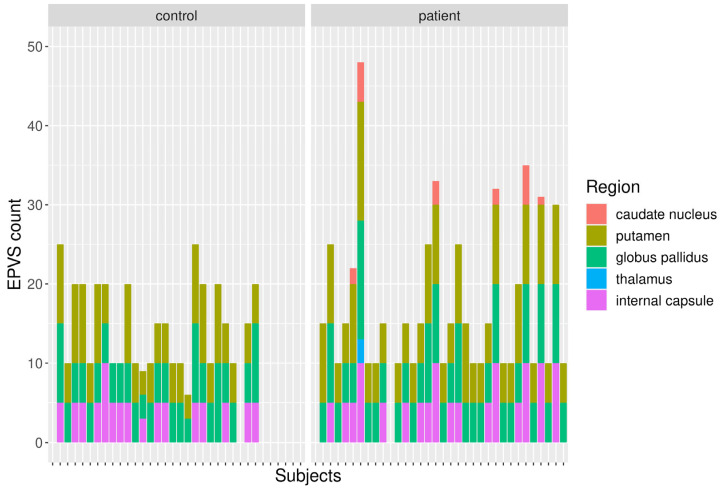
Region-wise EPVS count in HC and patients.

**Figure 5 biomedicines-13-02937-f005:**
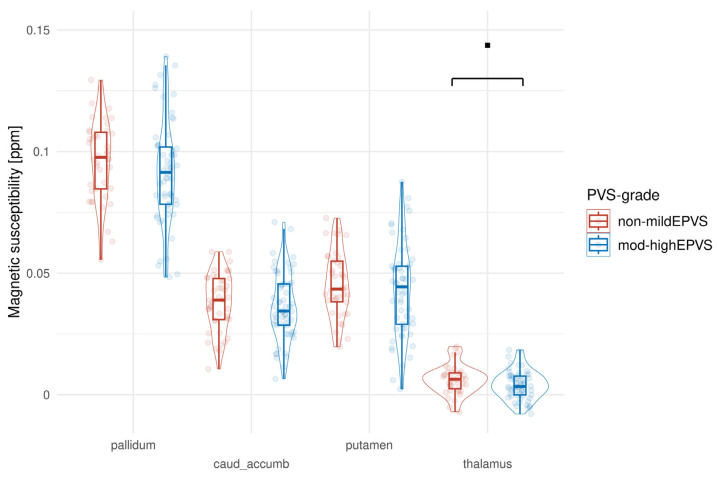
Effect of EPVS on regional susceptibility in patients and controls. Boxes represent the interquartile range with the median value displayed as a line within the box. Non-significant trend in magnetic susceptibility is indicated by ▪ (*p* = 0.08).

**Figure 6 biomedicines-13-02937-f006:**
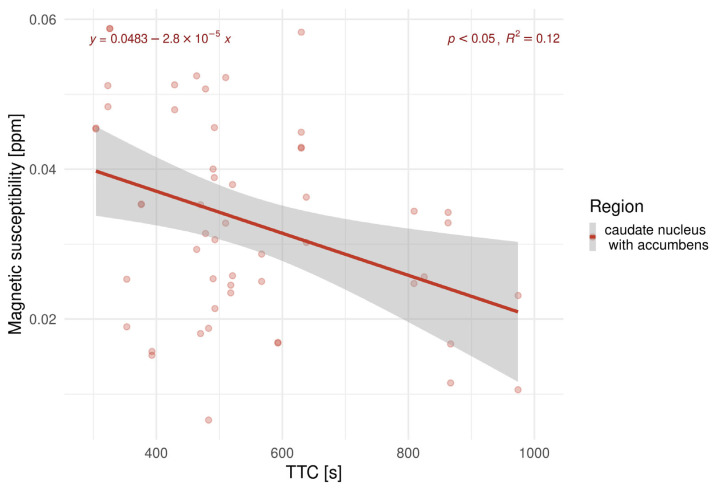
Linear regression analyses examining the relationship between TTC and susceptibility in the caudate nucleus with the accumbens of HE patients.

**Figure 7 biomedicines-13-02937-f007:**
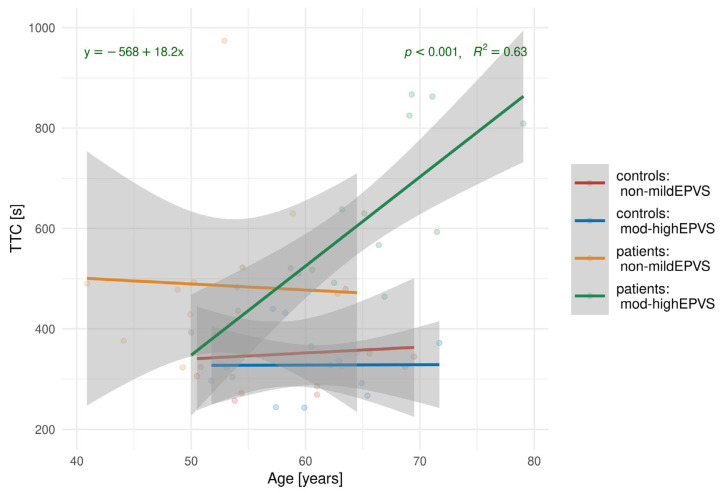
Linear regression analyses examining the relationship between TTC and age, stratified by control and patient groups and EPVS subgroups.

**Table 1 biomedicines-13-02937-t001:** Comparison between controls and patients in demographics and Mini-Mental State Examination (MMSE) score, Time To Complete (TTC), enlarged perivascular space (EPVS) grade, Child-Pugh rate, and HE subgroups.

Variable	*N*	Control [*N* = 25] ^1^	Patient [*N* = 28] ^1^	*p*-Value ^2^
*Gender*	53			0.14
female		11/25 (44%)	7/28 (25%)	
male		14/25 (56%)	21/28 (75%)	
*Age [years]*	53	60.1 (6.3)	59.5 (9.1)	0.7
*MMSE*	53	29.64 (0.70)	29.29 (1.24)	0.2
*TTC [s]*	52	338 (76)	548 (178)	<0.001
*EPVS*	53			>0.9
non-mildEPVS		11/25 (44%)	12/28 (43%)	
mod-highEPVS		14/25 (56%)	16/28 (57%)	
*CHILD rate*	28	NA (NA)	6 (3)	
*HEsubgroup*	28			>0.9
nonHE			11/28 (39%)	
minHE			17/28 (61%)	

^1^ n/N (%); Mean (SD). ^2^ Pearson’s Chi-squared test; Wilcoxon rank sum test; Fisher’s exact test.

## Data Availability

Fully anonymized data will be made available on request.

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
