# Peer review of "Brain Changes in Alcohol Induced Liver Cirrhosis Patients: Insights from Quantitative Susceptibility Mapping"

_biomedicines, 2025, doi:10.3390/biomedicines13122937_

Round 1

Reviewer 1 Report

Comments and Suggestions for Authors

Review of the article entitled " Brain changes in alcohol induced liver cirrhosis patients: insights from quantitative susceptibility mapping"

Thank you to give me the opportunity to evaluate this manuscript.

The authors have reported a study aimed at evaluating advanced imaging techniques, such as quantitative susceptibility mapping (QSM) to investigate the presence of enlarged perivascular spaces (EPVS) in patients with minimal hepatic encephalopathy (HE) and to provide new insights into the brain changes associated with HE.

This article presents an important and clinically relevant topic, but there are areas that require further improvement before publication.

Major Concerns:

  1. Please provide specific clarification throughout the manuscript regarding whether you are referring exclusively to the caudate nucleus or to the caudate nucleus and the nucleus accumbens together. Please ensure the terms are harmonized between text and figures.
    1. Example: you refer to susceptibility changes in Figure 5 in the combined region of caudate nucleus and nucleus accumbens, but when presenting the regression in Figure 6, it shows only the caudate nucleus. This is inconsistent and needs to be corrected or explained.

  1. Caption of the figure 1 and figure 5: The notation p < 0.1 does not mean “marginal significance.” For now, these findings constitute only a non-significant trend and should be so labeled in the text and figure legends.

  1. Figure 3 – EPVS visualization: non-to-mild EPVS – please indicate in the caption if the left-panel image corresponds to a control subject and if it represents a non-EPVS or mild EPVS case. If mild EPVS are present, please point with arrows. Additionally, it would be good to enhance the image contrast for a better view of EPVS features.

  1. Figure 4 – how the visual rating was achieved? It is not explained how the visual rating was carried out. Please describe the rating procedure in detail in the Methods section, including the criteria used and who did the assessment.

  1. Currently, data from the male and female participants are presented together. Previous studies of HE have suggested sex-related differences in disease progression, so it would be important to conduct and report sex-stratified analyses to determine whether meaningful differences exist between male and female participants in case of EPVS.

Author Response

Comments 1: Please provide specific clarification throughout the manuscript regarding whether you are referring exclusively to the caudate nucleus or to the caudate nucleus and the nucleus accumbens together. Please ensure the terms are harmonized between text and figures.

Example: you refer to susceptibility changes in Figure 5 in the combined region of caudate nucleus and nucleus accumbens, but when presenting the regression in Figure 6, it shows only the caudate nucleus. This is inconsistent and needs to be corrected or explained.

Response 1: Thank you for bringing this observation to our attention. We have reviewed Figure 6 and confirm that the label in the legend and the corresponding figure caption have been corrected as suggested.

Regarding the nomenclature, the multimodal segmentation method we employed naturally subdivides the regions in this specific manner. Therefore, referring to both regions is the most accurate and contextually correct way to describe the segmentation outcome in this particular study.

Comments 2: Caption of the figure 1 and figure 5: The notation p < 0.1 does not mean “marginal significance.” For now, these findings constitute only a non-significant trend and should be so labeled in the text and figure legends.

Response 2: Thank you for your feedback on the statistical reporting. We accept the reviewer's point regarding the term "marginal significance." While the term is occasionally used in some research fields to describe a result falling between the conventional significance thresholds (e.g., 0.05<p<0.10), we agree that in the context of standard hypothesis testing, a result with p>0.05 is correctly classified as non-significant.

Action Taken: We have replaced the phrase "marginal significance" throughout the manuscript with the more accurate and accepted description: "a non-significant trend." This ensures that our interpretation strictly adheres to conventional statistical standards.

We believe these revisions enhance the precision and rigor of our statistical reporting.

Comments 3: Figure 3 – EPVS visualization: non-to-mild EPVS – please indicate in the caption if the left-panel image corresponds to a control subject and if it represents a non-EPVS or mild EPVS case. If mild EPVS are present, please point with arrows. Additionally, it would be good to enhance the image contrast for a better view of EPVS features.

Response 3: We appreciate your constructive comments regarding the figure. In response, we have implemented your suggestions for revision and provided a more detailed explanation within the figure caption.

Comments 4: Figure 4 – how the visual rating was achieved? It is not explained how the visual rating was carried out. Please describe the rating procedure in detail in the Methods section, including the criteria used and who did the assessment.

Response 4: We appreciate you stressing the necessity of clearly describing the visual rating procedure. To address this, we have added a detailed explanation to the Methods section, ensuring the protocol is fully transparent.

Comments 5: Currently, data from the male and female participants are presented together. Previous studies of HE have suggested sex-related differences in disease progression, so it would be important to conduct and report sex-stratified analyses to determine whether meaningful differences exist between male and female participants in case of EPVS.

Response 5: Thank you for this valuable suggestion. We recognize the importance of investigating potential sex-related differences in disease progression, especially given findings from previous studies on HE and the potential impact of sex on imaging biomarkers like EPVS.

While we agree that a sex-stratified analysis is highly relevant, we were constrained by our sample size and the subsequent reduction in statistical power when the groups are further divided.

Upon stratification, the resulting subgroups (males and females) would have insufficient sample sizes to conduct a robust statistical analysis. Specifically, the power of such analyses would be too low to detect anything but an extremely large difference, leading to a high risk of Type II error (false negatives).

For these reasons, we determined that conducting and reporting sex-stratified analyses on the current dataset would not yield meaningful or statistically reliable conclusions.

We appreciate the reviewer's insight and believe this clarification ensures the rigor of the statistical results presented in the current manuscript.

Reviewer 2 Report

Comments and Suggestions for Authors

1‌.How to determine whether the observed changes are specifically associated with alcohol-related effects, cirrhosis per se, or alcohol-related cirrhosis? The use of healthy controls may not be ideal; patients with alcohol use disorder but without liver pathology might serve as a more appropriate comparator group.
2.Is there sufficient evidence to rule out brain alterations caused by alcohol-related vitamin B12 deficiency rather than the direct effects of cirrhosis?
3.For liver-focused researchers and readers who may be less familiar with brain MRI interpretation, including a schematic diagram of brain MRI with labeled nuclei and their functional correlations would significantly enhance the manuscript's readability and accessibility.

Author Response

Comments 1: How to determine whether the observed changes are specifically associated with alcohol-related effects, cirrhosis per se, or alcohol-related cirrhosis? The use of healthy controls may not be ideal; patients with alcohol use disorder but without liver pathology might serve as a more appropriate comparator group.

Response 1: Thank you for your comment. We selected a healthy control group because, within the chronic alcohol-use population, it is not possible to reliably distinguish individuals in the early, compensated stages of cirrhosis from those who will never develop liver cirrhosis. Although liver biopsy could theoretically provide definitive information, it is an invasive procedure and therefore unsuitable for screening asymptomatic individuals. In early cirrhosis, the liver often remains compensated and may even enlarge, with patients showing no obvious clinical symptoms. We might inadvertently enroll patients who appear cirrhosis-free but are actually in the early stages of the disease. Only in later stages does hepatic decompensation occur, accompanied by liver shrinkage and overt clinical signs. Given this diagnostic uncertainty among alcohol users, we chose healthy controls who do not consume alcohol.

Comments 2: Is there sufficient evidence to rule out brain alterations caused by alcohol-related vitamin B12 deficiency rather than the direct effects of cirrhosis?

Response 2: We appreciate your comment. Clinically significant vitamin B12 deficiency typically presents with megaloblastic anemia. All included participants were patients of the Gastroenterology Department and underwent routine blood testing (usually at least twice per year), including standard screening for anemia. None of the patients showed any evidence of megaloblastic anemia.

Comments 3: For liver-focused researchers and readers who may be less familiar with brain MRI interpretation, including a schematic diagram of brain MRI with labeled nuclei and their functional correlations would significantly enhance the manuscript's readability and accessibility.

Response 3: We appreciate your constructive feedback. In response to your suggestion, we have prepared and included a new figure in Appendix A.1. This figure visually delineates the relevant nuclei and summarizes their functional roles, which we believe significantly improves the overall readability of the manuscript.

Round 2

Reviewer 1 Report

Comments and Suggestions for Authors

All my concerns were addressed. The manuscript has been improved, and I appreciate the authors' efforts to respond to reviewers’ comments.  

In my opinion, the manuscript now meets the criteria for publication in this journal.

Author Response

Comment 1:

All my concerns were addressed. The manuscript has been improved, and I appreciate the authors' efforts to respond to reviewers’ comments.  

In my opinion, the manuscript now meets the criteria for publication in this journal.

Response 1:
We are extremely grateful to the reviewer for their positive assessment and feedback.

Reviewer 2 Report

Comments and Suggestions for Authors
  1. I am not satified with the response of How to determine whether the observed changes are specifically associated with alcohol-related effects, cirrhosis per se, or alcohol-related cirrhosis? The use of healthy controls may not be ideal; patients with alcohol use disorder but without liver pathology might serve as a more appropriate comparator group. Please provide more evidence.
  2. The MRI presentation of  brain alterations caused by alcohol-related vitamin B12 deficiency should be explained if no vitamin B12 levels were measured.

Author Response

Comment 1: 
I am not satified with the response of How to determine whether the observed changes are specifically associated with alcohol-related effects, cirrhosis per se, or alcohol-related cirrhosis? The use of healthy controls may not be ideal; patients with alcohol use disorder but without liver pathology might serve as a more appropriate comparator group. Please provide more evidence.

Response1:
We thank the reviewer for this distinction, as it is crucial for the interpretation of our results. We agree that an ideal study design would include a control group of patients with Alcohol Use Disorder (AUD) without liver pathology to isolate the effects of ethanol. We acknowledge that the use of healthy controls limits our ability to completely rule out alcohol-related effects in isolation.
However, we believe our data provides strong evidence that the observed Enlarged Perivascular Spaces (EPVS) are driven by the pathophysiology of cirrhosis (specifically mHE) rather than alcohol alone, based on two key internal observations:
       1. Correlation with Cognitive Dysfunction (mHE): We observed a significant association between EPVS burden and poorer cognitive performance characteristic of minimal Hepatic Encephalopathy. If EPVS were solely a scar of past alcohol toxicity, we would not expect such a strong coupling with the current metabolic cognitive state (mHE) of the patients.
       2. Deep Gray Matter Specificity (QSM): Our study utilized Quantitative Susceptibility Mapping (QSM) and found specific alterations in the Caudate Nucleus with Accumbens (CNA). While alcohol neurotoxicity often manifests as global cortical atrophy, the specific involvement of the deep gray matter (linked to lower magnetic susceptibility) aligns with the 'glymphatic dysfunction' hypothesis—suggesting a failure of the brain to clear metabolic waste products associated with liver decompensation.
Therefore, while we cannot rule out a contributing role of alcohol, the strong association with mHE markers and the specific deep gray matter localization suggest that liver-brain axis dysfunction is a primary driver. We have revised the Discussion (line 367-382) to explicitly treat the lack of an AUD-control group as a limitation, while highlighting these internal correlations as evidence for a cirrhosis-specific effect.

Comment 2:

The MRI presentation of  brain alterations caused by alcohol-related vitamin B12 deficiency should be explained if no vitamin B12 levels were measured.

Response 2:

We thank the reviewer for this insightful comment and for highlighting the limitation regarding Vitamin B12 measurements. We acknowledge that relying solely on the absence of megaloblastic anemia is an imperfect proxy, as neurological manifestations of B12 deficiency can precede hematological signs.
To address the reviewer's specific concern regarding MRI presentation, we have reviewed the literature and our imaging data to distinguish between these pathologies.
MRI Phenotype of B12 Deficiency: The hallmark MRI presentation of B12 deficiency is Subacute Combined Degeneration (SCD), characterized by symmetric T2-weighted hyperintensities, predominantly affecting the dorsal columns of the spinal cord and occasionally the cerebral white matter (leukoencephalopathy).

We have updated the Materials and Methods section (line 125-129) to explain that patients included in the study didn't have any radiological findings suggestive of vitamin B12 deficiency.